# Kinin Receptors and Kinin-Related Gene Expression in Astrocytic Brain Tumors

**DOI:** 10.3390/cancers16020241

**Published:** 2024-01-05

**Authors:** Izabela Stadnicka, Barbara Strzałka-Mrozik, Magdalena Kimsa-Dudek, Wojciech Kaspera, Andrzej Plewka, Wojciech Szopa, Antoni Stadnicki

**Affiliations:** 1Department of Molecular Biology, Faculty of Pharmaceutical Sciences in Sosnowiec, Medical University of Silesia, 40-055 Katowice, Poland; izabela.stadnicka1@gmail.com; 2Department of Nutrigenomics and Bromatology, Faculty of Pharmaceutical Sciences in Sosnowiec, Medical University of Silesia, 40-055 Katowice, Poland; mkimsa@sum.edu.pl; 3Department of Neurosurgery, Medical University of Silesia, St. Barbara Hospital, 41-200 Sosnowiec, Poland; wkaspera@sum.edu.pl (W.K.); wszopa@sum.edu.pl (W.S.); 4Institute of Health Sciences, University of Opole, 45-040 Opole, Poland; plewka.andrzej@gmail.com; 5Faculty of Medicine, Jan Długosz University in Częstochowa, 42-200 Częstochowa, Poland; astadnic@wp.pl; 6Section of Gastroenterology, Multidisciplinary Hospital, 43-600 Jaworzno, Poland

**Keywords:** kinin receptors, bradykinin, gliomas, oligonucleotide microarrays, RT-qPCR

## Abstract

**Simple Summary:**

Kinin receptors have been implicated in cancer migration, invasion, angiogenesis, and metastasis. However, research on the molecular changes that occur during glioma development remains incomplete. Therefore, we assessed differences in the expression of kinin receptors and kinin-dependent genes in forty-three patients with astrocytic gliomas of various tumor grades (G2, G3, and G4). In our study, we confirmed that kinins and their receptors are involved in the development of glioma. We also found genes whose expression change may have diagnostic and therapeutic significance.

**Abstract:**

Kinins are a set of peptides present in tissues that are involved in the inflammatory response and cancer progression. However, studies showing the expression of kinin receptors in human glioma samples are still incomplete and contradictory. The aim of the present study was to ascertain the expression of *BDKRB1* and *BDKRB2* genes, as well as the level of B1R and B2R proteins in human gliomas, depending on the degree of malignancy. Additionally, representative kinin-dependent genes with altered expression were indicated. The expression profile of kinin-dependent genes was determined using oligonucleotide microarray technique. In addition, RT-qPCR was used to assess the expression level of selected differentiating genes. The location of kinin receptors in brain gliomas was assessed using immunohistochemical methods. The oligonucleotide microarray method was used to identify 12 mRNA IDs of kinin-related genes whose expression was upregulated or downregulated in gliomas of different grades. In immunohistochemically stained samples, the concentrations of BR1 and BR2 proteins, measured by optical density, were statistically significantly higher in grade G3 vs. G2 and G4 vs. G3. Increased expression of kinin receptors *BDKRB1* and *BDKRB2* in brain gliomas, depending on the degree of malignancy, suggests the involvement of kinins and their receptors in the disease’s pathogenesis. Quantitative assessment of mRNA *BDKRB1*, *PRKAR1A*, *MAP2K*, and *EGFR* in patients with brain tumors may hold diagnostic and therapeutic significance.

## 1. Introduction

Kinins are bioactive polypeptides that are physiologically present in body fluids and many tissues, where they participate in the regulation of blood pressure, glucose and electrolyte transport, cell proliferation, leukocyte activation, cell migration, endothelial cell activation, and nociception [1].

Two kallikreins, plasma and tissue, cleave kininogens to release kinins. Plasma kallikrein releases bradykinin (BK) from high-molecular-weight kininogen, while tissue kallikrein liberates Lys-BK from low-molecular-weight kininogen [2]. The pleiotropic functions of kinins depend on signaling through two G-protein-coupled cross-talking receptors: the constitutively expressed kinin receptor 2 (B2R) and the inducible kinin receptor 1 (B1R). There is little difference in the biological activity of BK and Lys-BK. Both are converted to desArg derivatives by carboxypeptidase N or M and become B1 receptor agonists [3]. However, dysregulation of the kinin system is involved in various pathological conditions. Both B1R and B2R may be upregulated following inflammatory stimuli, cancerous proliferation, and traumatic conditions [3,4,5]. Previous studies revealed that B1R and B2R can be upregulated in various human cancers, among others [6,7]. Activation of kinin receptors and their cross-talk with receptor tyrosine kinases activate multiple signaling pathways, including ERK/MAPK, PI3K, PKC, and p38 pathways that regulate cancer hallmarks. Perturbations in kinin-mediated events are implicated in various aspects of cancer invasion, matrix remodeling, metastasis, proliferation of tumor cells, and inhibition of apoptosis. In the tumor microenvironment, kinins initiate fibroblast activation, interactions with mesenchymal stem cells, and the recruitment of immune cells. Although the precise nature of kinin function in metastasis and the tumor microenvironment is not completely clear yet, several kinin receptor antagonists show anti-metastatic potential [8].

Often, cross-talk between B1R and B2R with epidermal growth factor receptor (EGFR) receptors, either via intracellular transactivation or by extracellular shedding of HER ligands, maintains tumor growth, as reported in glioblastoma (GBM) [9].

Furthermore, kinin receptors, as endogenous algogenic mediators associated with various painful conditions via B1R and B2R receptor activation, have also been implicated in cancer-related pain [10].

Production of bradykinin and expression of kinin receptors, resulting from oxygen deficiency in a fast-growing tumor, drive an autocrine loop leading to tumor growth [11]. Additionally, bradykinin stimulates the angiogenesis process by activating vascular endothelial growth factor (VEGF) and increases tissue permeability by activating metalloproteinases [8]. Moreover, previous studies have revealed an increased number of bradykinin receptors in cells transformed by the activation of oncogenes [11]. B2R overexpression is oncogenic, and other regulators influence its gene expression. Overexpression of ras gene products (Ha-, K-, and N-), both normal and transformed forms, influenced B2R expression. B2R level is upregulated in all three p21ras overexpressed cells, leading to increased sensitivity of BK-B2R binding and DNA synthesis [6].

In the central nervous system, kinins, which act through B2R and B1R, may stimulate other pro-inflammatory mediators and are considered to play a role in neurodegenerative disorders [12,13]. Importantly, kinins have recently been described as mediators associated with brain tumor progression [9,14]. In glioblastoma (GBM) cells, the BK-B1R pathway mediated overexpression of aquaporin-4 (AQP4) and resulted in heightened migration and invasion. AQP-4 changes the entire tumor cell surface cytoskeletal structures by disturbing the arrangement of orthogonal arrays of particles [15].

Gliomas are the most common primary tumors of the central nervous system. According to the latest 2021 World Health Organization (WHO) classification of tumors of the central nervous system (CNS), adult-type diffuse gliomas are divided based on histological features and isocitrate dehydrogenase (IDH) mutation status. IDH wild-type astrocytomas with histological features of malignancy such as mitotic activity, necrosis, and microvascular proliferation are diagnosed as glioblastoma (GBM) grade 4 (G4). Additionally, all diffuse astrocytic tumors with IDH mutations are classified as grade 2 (G2), grade 3 (G3), or grade 4 (G4) based on histological features. These tumors are considered oligodendrogliomas when a 1p/19q codeletion is found [16].

In vitro studies have indicated that BK activation of B2R can induce the migration of astrocytic and microglial cells [17,18]. Furthermore, BK can mediate the migration of glioma cells toward blood vessels through the activation of B2R [19]. Other in vitro studies have shown that kinins, through the activation of B1R, can stimulate the migration of C6 rat glioma cells and U-251MG human glioblastoma cells, as well as induce COX-2 expression [20]. Recently, Nikotetti et al. [21] reported that the stimulation of B1R and B2R with des-Arg 9 -BK and BK, respectively, induced the proliferation of human glioma cells U-138MG and U-251MG. Conversely, their inhibition with selective kinin receptor antagonists mediated glioma cell death [21]. Pillat et al. [22] proposed a different approach to glioma cell invasion, suggesting that BK decreases B1R and B2R expression in mesenchymal stem cells (MSCs) while increasing their expression in the GBM cell line. This indicates that BK may influence the cross-talk between GBM and MSCs, which is significant for tumor progression [22]. Furthermore, activation of kinin receptors leads to the activation of macrophages, astrocytic cells, and matrix metalloproteinases (MMPs) in the tumor environment [14,23]. In an experimental model of neuroblastoma, Naido and Raido [24] demonstrated that the expression of kinin receptors, through the activation of MMPs, leads to the degradation of the extracellular matrix in the tumor area, facilitating angiogenesis and tumor infiltration [24]. Collectively, in vitro results and experimental studies suggest that the activation of B1R and B2R may contribute to glioma progression.

In human studies, Raido et al. [25] assessed the localization of tissue kallikrein protein and kinin receptor proteins in resected astrocytic brain tumors using immunohistochemical methods. In unchanged brain tissue, immunoreactive B2R was found in cortical neurons but not in glial cells. In human glioma tissues, both B2R and B1R have been shown to be immunoreactive in astrocytic cells. However, in their examination of 12 cases, these authors did not report a dependence of the reaction intensity on the histopathological degree of glioma malignancy. In later studies, Zhao et al. [26] demonstrated an increase in B2R protein levels in accordance with the increased degree of glioma malignancy. In contrast, Nicoletti et al. [9] found significantly higher B1R expression in low-grade gliomas (G2 grade) compared to glioblastomas (G4 grade). Additionally, the expression of B2R proteins in gliomas was similar regardless of the degree of malignancy.

Previous studies that examined the expression of kinin receptors in human glioma samples, primarily utilizing immunohistochemical methods, have yielded incomplete and sometimes contradictory results.

The aim of this study was to determine the expression of *BDKRB1* and *BDKRB2* genes and the levels of B1R and B2R proteins in human gliomas, with consideration of the degree of malignancy. Additionally, we identified kinin-dependent genes with altered expression patterns, which could potentially aid in the diagnostic process and serve as crucial components in personalized treatment.

## 2. Materials and Methods

### 2.1. Patients

Forty-three consecutive patients with astrocytic glioma (22 men and 21 women, mean age 54 ± 14.0) were enrolled in this prospective study. The diagnosis and surgical resection of brain tumors were performed at the Department of Neurosurgery, Silesian Medical University, St. Barbara Hospital, in Sosnowiec. This study was conducted in accordance with the protocol approved by the Ethics Committee of the Silesian Medical University (Approval No. KNW/0022/KB1/59/13, year 2018), and all patients provided written informed consent.

Prior to study enrollment, patients initially received a diagnosis of brain glioma through contrast-enhanced computed tomography (CT) scans. Subsequently, magnetic resonance imaging (MRI) of the brain was conducted, including T1- and T2-weighted sequences, FLAIR sequence (a modification of the T2-weighted sequence), and/or diffusion imaging (diffusion tensor—DWI). Patients with NYHA III and IV degrees of circulatory failure and those taking medications such as angiotensin-converting enzyme (ACE) inhibitors and sartans, which may affect the kinin system, were excluded from the study.

Following surgical resection, brain tumor biopsy specimens were obtained for histopathological examination and molecular analysis. Tissue samples for molecular examination were promptly preserved in RNAlater^®^ (Sigma-Aldrich, St. Louis, MO, USA) at −20 °C for 24 h until RNA extraction.

The definitive diagnosis was established through histopathological evaluation of the resected lesion. Among the patients, twelve were diagnosed with grade 2 (G2) tumors, five with grade 3 (G3), and twenty-six with grade 4 (G4).

### 2.2. Extraction of Ribonucleic Acid (RNA) from Tissue Specimens

Brain tumor fragments from astrocytic sections were homogenized using a Polytron^®^ homogenizer (Kinematics AG, Uster, Switzerland). Total RNA extraction was carried out from the homogenized samples using TRI Reagent (Sigma-Aldrich, St. Louis, MO, USA) following the manufacturer’s instructions. Purification was achieved through the RNeasy Mini Kit (Qiagen Inc., Hilden, Germany) with columns and DNase I treatment (RNase-Free DNase Set, Qiagen Inc., Hilden, Germany).

Quantitative evaluation of the obtained total RNA was performed by the spectrophotometric technique using a MaestroNano MN-913 nanospectrophotometer (MaestroGen Inc., Las Vegas, NV, USA), with the assumption that the RNA solution had an OD of 1 corresponding to 40 µg/mL. The extracted material displayed satisfactory purity, as indicated by A260/A280 ratios falling between 1.8 and 2.0. To qualitatively assess the total RNA, we utilized horizontal electrophoresis with a 1% agarose gel stained with Simply Safe reagent (EurX, Gdańsk, Poland).

### 2.3. Oligonucleotide Microarray Analysis

Ten brain tumor section samples were selected for preliminary analysis on oligonucleotide microarrays, consisting of 4 samples from grade 2 (G2) and 6 samples from grades 3 and 4 (G3/G4). The kinin-related gene expression profile was assessed using HG-U133A 2.0 plates from Affymetrix (Santa Clara, CA, USA), following the manufacturer’s protocol. The procedure for transcriptome analysis with this oligonucleotide microarray has been previously described [27].

### 2.4. Reverse Transcription Quantitative Polymerase Chain Reaction

The microarray experiment was validated through real-time quantitative reverse transcription polymerase chain reaction (RT-qPCR) on all 43 brain tumor section samples. Each sample underwent three replicates of testing.

Quantitative analysis was performed using the Opticon™ DNA Engine Sequence Detector (MJ Research Inc., Watertown, MA, USA), relying on fluorescence measurements for a known quantity of cDNA copies. This study utilized sense and antisense primers with sequences that complemented the target genes’ sequences (Table 1), along with the Sensi-Fast^TM^ reagent kit (Bioline, London, UK). The β-actin gene (*ACTB*) was employed as a positive control for amplification. The thermal cycling conditions for the one-step RT-qPCR were as follows: reverse transcription at 45 °C, polymerase activation at 95 °C for 2 min, followed by 40 duplicate cycles comprising denaturation at 95 °C for 5 s and annealing at 60 °C for 10 s, concluded by a final elongation at 72 °C for 5 s. The mRNA copy number was determined using the standard curve method, as previously described by Strzałka-Mrozik et al. [28]. The specificity of the RT-qPCR reaction was evaluated by determining the melting temperature (Tm) of the amplimers and confirmed by agarose gel electrophoresis. The final expression levels of the examined genes were quantified as the number of mRNA copies per 1 μg of total RNA.

### 2.5. Immunolocalization of Kinin B1 and B2 Receptors in Astrocytic Brain Tumors

The localization of kinin receptors (B1R and B2R) was determined through immunohistochemistry in brain tumor specimens collected after surgical resection. Glioma tissue samples were subjected to dewaxing in xylene and hydration using a series of decreasing alcohol concentrations. To retrieve the antigen, we employed Target Retrieval Solution (TRS) from Dako (Carpinteria, CA, USA) heated to 95–97 °C in a water bath, followed by cooling to room temperature and rinsing in TRIS-buffered Saline (TBS, pH 6.7) (Dako, Carpinteria, CA, USA).

Next, to inhibit endogenous peroxidase activity, the tissue sections were exposed to a 3% hydrogen peroxide solution at room temperature for 5 min. To assess the concentration and localization of B1R and B2R proteins, the sections were incubated for 30 min at room temperature with rabbit polyclonal antibodies specifically targeting B1R (mix AS434) and B2R (mix AS287-83), respectively (provided by Prof. Dr. W. Muller-Ester, Institute of Biochemistry II, University of Mainz, Mainz, Germany). These antibodies were used at a concentration of 200 μg/mL, and a dilution of 1:100 was applied for each antibody. As a control, normal rabbit serum was used in place of anti-B2R or anti-B1R antibodies.

The subsequent steps of the immunohistochemical procedure were conducted using the standard Vector Laboratories ImmPRESS™ HRP Anti-Rabbit IgG (Peroxidase) Polymer Detection Kit (Vector Laboratories, Newark, CA, USA). The chromogen DAB (Dako, Carpinteria, CA, USA) was utilized to generate the color reaction. Finally, the sections were counterstained with hematoxylin, dehydrated through a series of alcohol solutions, cleared in xylene, and sealed with DPX mounting medium. An impartial observer then analyzed the sections. To evaluate B1R and B2R protein concentrations in the tumor samples, the optical density of the reaction products was measured using Image Pro Plus 6.0 software. This measurement was performed using an Olympus Provis AX 70 light microscope equipped with an Olympus DP 10 camera with a magnification of 200×.

Optical density was assessed in five randomly chosen fields within each specimen. The results were presented as the mean optical density (OD) with standard deviation (SD). The reaction was observed in both the cell membrane and cytoplasm of glioma cells and endothelial cells.

### 2.6. Statistical Analysis

The results obtained from the oligonucleotide microarray analysis were analyzed using the PL-Grid Infrastructure (http://www.plgrid.pl/ (accessed on 16 September 2022)) and the GeneSpring 13.0 platform by Agilent Technologies (UK Limited, South Queensferry, UK). Significant changes in fluorescence signals between the compared samples, expressed as fold change (FC), were considered when the absolute value exceeded 2.0, indicating at least a 2.0-fold increase or decrease in signal intensity. Statistical significance was set at *p* < 0.05.

For the analysis of mRNA copy numbers obtained via RT-qPCR, Microsoft Office Professional Plus 2016 and Excel were used. Subsequently, a database was created with Statistica software (version 13.1) from StatSoft, Tulsa, OK, USA, for further statistical analysis. The normality of data distribution was assessed using the Shapiro–Wilk test.

Since the data did not follow a normal distribution, the Kruskal–Wallis test, a non-parametric alternative to one-way analysis of variance (ANOVA), was employed to assess differences among the study groups. Further investigation of specific relationships was conducted using a post hoc test for multiple comparisons of mean ranks. Descriptive statistics, including the median (Me), minimum, and maximum values, as well as upper and lower quantiles, were also calculated. A significance level of *p* < 0.05 was applied in all statistical analyses.

For the analysis of protein concentrations, differences between groups in relation to individual variables were initially assessed using one-way ANOVA. Subsequent analysis of specific relationships was performed using Tukey’s post hoc test.

## 3. Results

### 3.1. Gene Expression Profile of Kinin-Related Genes Based on Oligonucleotide Microarray

The panel of 120 mRNA IDs of kinin-dependent genes, selected from a pool of 22,283 mRNA IDs based on the Affymetrix database and data in the literature, underwent comparative analysis. This comparative analysis was initiated with an assessment of the coherence in grouping the studied transcriptomes using the hierarchical clustering method. Through cluster analysis involving the fluorescence signals of 22,283 mRNA IDs on the HG-U133A 2.0 plate, two distinct groups of transcriptomes were identified, demonstrating similar expression patterns: grade 2 gliomas (referred to as low-grade gliomas, G2) and grade 3 and 4 gliomas (termed high-grade gliomas, G3/G4). This categorization served as the basis for subsequent microarray analyses.

In the subsequent phase, a heatmap was generated to visually represent the extent of variation between the compared transcriptome groups for the 120 mRNA IDs associated with kinin-related genes (Figure 1). The range of colors within the analyzed transcriptome groups signifies disparities in the expression profiles of kinin-related genes based on the tumor grade. To identify the differentially expressed kinin-related genes, a t-test was conducted. Following the analysis, a volcano plot illustrating the distinct genes between the studied groups was created.

Comparing the transcriptomes of glioma cells at various stages of advancement revealed statistically significant alterations in the expression of 12 mRNA IDs (Table 2), comprising three transcripts that were upregulated and nine that were downregulated (*t*-test, *p* < 0.05, FC > 2.0) (Table 3).

### 3.2. Expression of mRNA in Gliomas of Various Malignancy Stages Based on RT-qPCR

Based on the results obtained from the oligonucleotide microarray analysis, the following genes were selected for further validation using an independent real-time RT-qPCR method: *MAP2K2*, *EGFR*, *SHC1*, *PRKCA*, *PRKAR1A*, *BDKRB1*, and *BDKRB2*. The expression levels of these tested genes, along with the expression of the endogenous control *ACTB*, were quantified as the number of mRNA copies per 1 μg of total RNA.

The expression of *ACTB* mRNA and the selected kinin-related genes was detected in all tested samples, regardless of the stage of the lesion. In patients with G4 grade glioma, the copy number of *BDKRB1* mRNA copies/μg RNA (Me = 890.10 copies/μg RNA) was significantly higher (*p* = 0.020) compared to the mRNA *BDKRB1* level in patients with G2 grade (Me = 425.13 copies/μg RNA) (Figure 2A). Similarly, in glioma patients with G4 grade, the *BDKRB2* expression level (Me = 19,977.61 copies/μg RNA) was significantly increased (*p* < 0.001) compared to G2 grade (Me = 3317.50 copies/μg RNA) (Figure 2B).

The average numbers of *PRKAR1A* mRNA copies in glioma G3 patients (Me = 602,700.25 copies/μg RNA), as well as in glioma G4 patients (Me = 548,360.50 copies/μg RNA), were significantly higher compared to G2 grade (Me = 327,519.69 copies/μg RNA) (*p* = 0.029 and *p* = 0.003, respectively) (Figure 3A). The *PRKCA* mRNA levels were increased in patients with glioma G4 grade (Me = 48,862.06 copies/μg RNA) compared to G2 grade (Me = 28,870.08 copies/μg RNA); however, the difference between G4 grade and G2 grade did not reach significance (*p* = 0.091). In contrast, in patients with G4 grade, the mean levels of *MAP2K2* mRNA (Me = 48,071.09 copies/μg RNA), as well as *EGFR* mRNA (Me = 29,540.14 copies/μg RNA), were significantly lower (*p* = 0.036 and *p* = 0.004, respectively) compared to the mean mRNA levels of glioma patients with G2 grade (Me = 111,980.19 and Me = 160,789.13 copies/μg RNA, respectively) (Figure 3B,C). The *SHC1* mRNA levels and *ACTB* mRNA in patients with G4 grade did not differ significantly from G2 grade.

### 3.3. Immunochemical Localization and Concentration of Kinin Receptor Proteins

Both B1 and B2 kinin receptors were also detected in various stages of astrocytic gliomas (Figure 4 and Figure 5, respectively). In grade G2, endothelial cells of glioma vessels expressed both kinin receptors. The positive reaction was observed in the cell membrane as well as in the cytoplasm. B2R expression shown in the endothelium was more intense compared to that seen in glioma cells (Figure 5A). A weak reaction was observed in some glioma cells (Figure 4A and Figure 5A). In grade G3, the immunoreactive staining for both BR1 and BR2 is much more pronounced compared to grade G2, especially intracellularly. For BR2 (Figure 5B), a less intense staining reaction was observed in vascular endothelium than in glioma cells, whereas for BR1 (Figure 4B), the positive reaction in endothelial cells and glioma cells showed similar intensity. In grade G4, the staining reaction is very strong. For BR2 (Figure 5C), it is demonstrated in membrane structures of glioblastoma cells and in the endothelium of infiltrating vessels. For BR1 (Figure 4C), the positive reaction is particularly strong in vessels forming angiogenesis, with a similar staining intensity in the cell membrane and cytoplasm. The concentration of BR1 and BR2 within gliomas was measured as the mean optical density. BR1 protein was significantly higher (*p* < 0.05) in group G3 compared to group G2 and was shown to be the highest in group G4 (G4 vs. G3, *p* < 0.05). Similarly, the concentration of BR2 protein in glioma tissues was found to be significantly different (*p* < 0.05) according to the degree of glioma malignancy (G2 < G3 < G4), as shown in Table 4.

## 4. Discussion

Some tumors can be uniquely identified based on molecular alterations, while for others, this information provides only ancillary details [27]. The most commonly occurring genetic disturbances in gliomas involve the deregulation of tyrosine kinase receptors, such as the epidermal growth factor receptor (EGFR) and platelet-derived growth factor receptor (PDGFR), and their aberrant signaling through the phosphatidylinositol-3 kinase (PI3K) signaling pathway and the mitogen-activated protein kinase (MAPK) pathways, ultimately leading to malignant transformation [29]. Other critical genetic aberrations in glial tumors are associated with the loss of apoptosis due to p53 gene mutations and the stimulation of angiogenesis through VEGF signaling [30].

Low-grade gliomas, based on their isocitrate dehydrogenase (IDH) mutation status, were typically categorized into two groups. The first group consists of gliomas that develop through an IDH mutation, followed by a p53 mutation. In contrast, poor-prognosis low-grade gliomas (lacking IDH mutation) and primary glioblastomas (GBMs) are characterized by EGFR amplification along with deletions/mutations of the phosphatase and tensin homolog tumor suppressor gene (PTEN) [29,31].

In the latest 2021 WHO classification of central nervous system (CNS) tumors, gliomas that exhibit IDH mutations are classified as diffuse low-grade gliomas (DLGGs). IDH mutations are observed in more than 80% of this type of tumor [16].

It should be emphasized that among patients affected by DLGGs, the presence of 1p/19q codeletion is significantly associated with extended survival and a longer time to malignant transformation (TMT) compared to cases without 1p/19q codeletion. Additionally, the extent of surgical resection (EOR) in DLGG patients has been established as one of the primary prognostic factors. However, its predictive value is substantially influenced by the presence of the 1p/19q codeletion [32].

Additionally, the methylation status of O(6)-methylguanine-DNA methyltransferase (MGMT) is predictive of outcomes in GBM patients treated with alkylating agent chemotherapy. Brain gliomas should also be characterized based on gene expression to determine which genes are amplified and/or mutated [33].

Kinins and their receptors appear to play a role in cancer-related processes [6]. Previous research has shown that both the B1 and B2 receptors can be upregulated in human cancers [34]. Therefore, in the present study, we initially focused on analyzing the expression patterns of genes and proteins related to kinin receptors in human astrocytic gliomas.

The microarray analysis revealed a statistically significant increase in *BDKRB1* expression in the G3/G4 grade compared to the G2 grade. Similarly, there was a significant increase in *BDKRB1* mRNA copy number in the G4 grade compared to the G2 grade. However, the transcriptional activity of *BDKRB2* was lower in the G3/4 grade than in the G2 grade, whereas the *BDKRB2* mRNA level was significantly higher in the G4 grade than in the G2 grade. The disparity in the transcriptional activity of *BDKRB1* and *BDKRB2* may be explained by the “cross-talk” between B2R and B1R, as previously demonstrated in a rat model of kindling-induced epilepsy [35]. Alternatively, it should be underlined that B2R is thought to be rapidly desensitized in a cell after internalization, but B1R is more stable and resistant to desensitization. B1R is rather redistributed to caveolae-related lipid rafts in the cell membrane [36,37]. Regarding the pattern of kinin receptor proteins in human gliomas, our findings are consistent with the results of Zhao et al. [26], who observed an increase in B2R protein levels that correlated with the degree of glioma malignancy. Furthermore, we extended their study by demonstrating an increased expression of B1R that corresponds to the malignancy grade, ranging from G2 to G4. Therefore, the increased expression of both B1R and B2R proteins in gliomas, depending on the degree of malignancy, suggests the involvement of both kinin receptors in neoplastic transformation.

In terms of kinin-related genes, we observed that the transcriptional activity of *PRKAR1A*, which encodes protein kinase A (PKA), is upregulated in grade G3-G4 gliomas. This finding was further validated by quantifying mRNA levels from 43 resected brain tumor samples using RT-qPCR, which confirmed the overexpression of the *PRKAR1A* gene (G4 vs. G3 vs. G2, *p* < 0.001).

Protein kinase A (PKA) is composed of two catalytic subunits and contains two regulatory subunits. The major regulatory component that controls most of PKA’s catalytic activity in response to cyclic AMP (cAMP) is the type 1 alpha regulatory subunit. It is worth noting that the B2 receptor (RB2) can activate adenylate cyclase through the Gs protein, leading to the synthesis of cAMP from ATP and the subsequent activation of PKA [38]. An early in vitro study demonstrated that RB2 signaling in A431 human epidermoid carcinoma cells stimulates the Gs-mediated cyclic AMP pathway [39]. Recently, the importance of PKA in the proliferation and invasion of glioblastoma cells has been highlighted. In vitro studies using malignant glioma cell lines have demonstrated that the EGFRv III mutant promotes glioblastoma growth and invasion through PKA-dependent phosphorylation of Dock180. Dock180 is a bipartite guanine nucleotide exchange factor that catalyzes the exchange of GDP for GTP by GTPases [40]. In another in vitro study on rat glioma C6 cell lines, Tanabe et al. [41] showed that cAMP/PKA activation leads to IL-1β-induced amplification and IL-6 synthesis by activating the kinase JAK2 and the STAT3 transcriptional factor pathways. These findings underscore the role of PKA in the regulation of signaling pathways that contribute to the growth and invasiveness of glioblastoma cells. However, until now, there have been no data indicating a direct role of PKA activation via the kinin receptor pathway in glioma cells or human glioma tissue. In the present study, the overexpression of *PRKAR1A*, related to kinin receptors and confirmed by the higher expression of mRNA *PRKAR1A* in G4 and G3 grades compared to G2 glioma grade, suggests that kinin receptors may, in part, mediate molecular pathways via cAMP-dependent PKA in high-malignancy gliomas.

In the current study, the *PRPCA* gene, which encodes kinase C (PKC), was shown to be overexpressed in high-grade gliomas (G3/G4) compared to the G2 grade. An increase in *PRPCA* mRNA in the G4 grade compared to G2 was also observed, although the difference between these groups did not reach statistical significance (*p* = 0.094). Hsieh et al. [42] demonstrated in vitro that the activation of B2R can induce mitogenic signaling pathways dependent on PKC-delta in astrocytes. This leads to the phosphorylation of extracellularly regulated kinases 1 and 2 (ERK1/2) and subsequent activation of the transcription factor Elk-1, which, in turn, induces MMP-9 transcription [42]. Although this in vitro data may not directly apply to glioma cell lines, it is known that activated astrocytes in the brain microenvironment can promote the invasion of neoplastic cells [23]. Moreover, it has been demonstrated that B1R activation of PKC isoforms leads to the phosphorylation of ERK1/2 in estrogen-sensitive breast cancer cells, accompanied by an increase in tumor cell proliferation [43]. The presented in vitro data suggest a potential role of kinins and their receptors in activating PKC in gliomas. The results of the current study, showing the overexpression of the PRPCA gene encoding PKC, provide genomic-level support for this observation.

Another gene that was found to be overexpressed in high-grade gliomas (G3/G4) compared to G2 was the *PIK3CB* gene, which encodes the phosphatidylinositol beta-kinase-3 catalytic subunit. The activation of PI-3 kinase signaling pathways is well-known in the progression of gliomas, and the relationship between the activation of this pathway through kinin receptors has also been demonstrated in experimental studies. Therefore, we did not validate the assessment of *PIK3CB* gene expression by RT-qPCR.

Indeed, the activation of both B2R and B1R can initiate signal transduction pathways through PKC, resulting in the synthesis of inositol-3-phosphate (IP-3) and diacylglycerol (DAG) [14]. In vitro studies have demonstrated the activation of the PI3K/Akt (protein kinase Akt) pathways in glioblastoma cells due to the activation of kinin receptors [20,21]. Activation of PI-3 kinase signaling pathways in gliomas leads to the activation of downstream effector molecules such as Akt and mTOR (the mammalian target of rapamycin), which promote proliferation and resistance to apoptosis [44]. Inactivation of PTEN in high-grade gliomas also stimulates the activation of the PI-3 kinase pathway [29,45,46].

In contrast, the transcriptional activity of the *MAP2K2* gene (mitogen-activated protein kinase 2), which encodes the tyrosine kinase receptor EGFR (epidermal growth factor receptor), and the *EGFR* gene itself were downregulated. Quantitative assessment of mRNA from 43 samples of astrocytic brain tumors confirmed these changes for *MAP2K2* (G2 vs. G4, *p* < 0.001), as well as for *EGFR* (G2 vs. G4, *p* < 0.001).

MAP kinases are induced in gliomas through multiple signaling pathways. The *MAP2K2* gene encodes a MAP2K2 kinase (mitogen-activated protein kinase 2) which phosphorylates and thus activates MAPK1/ERK2 and MAPK2/ERK3. Genetic alterations in pathways affect the MAP kinases network, including RAS (protein-regulated signal pathways), in the MAP/ERK, c-Jun N-terminal kinase (JNK, stress-activated protein kinases), and MAPp38 pathways, leading to malignant transformation [47]. Regarding the role of kinins and kinin receptors in the MAP signaling cascade, it was demonstrated in rat astrocyte cell lines that stimulation with BK caused activation of the p42/p44 MAPK kinase and Akt signaling pathways, dependent on the nuclear factor NF-κβ [48]. In another in vitro study by Nicoletti et al. [21], conducted on the human glioblastoma cell lines U-138MG and U-251MG, it was shown that the stimulation of B1R and B2R through kinin receptor agonists activates the PI-3/Akt kinase and ERK kinase pathways, leading to the proliferation of glioblastoma cells.

It is worth noting that previous studies have predominantly focused on genetic abnormalities in higher-grade gliomas (G3 and G4, according to WHO). Comparatively, there has been significantly less attention given to the molecular biology of WHO G2 grade gliomas. Recent research suggests that mutations and overexpression of EGFR in both pediatric and adult low-grade gliomas (G2) can lead to the activation of the RAS/MAPK kinase pathway, driving the proliferation of glioma cells [49,50]. Furthermore, genetic alterations in EGFR of glioma G2 grade lead to abnormal mitogen-activated protein kinase (MAP kinase) pathway signaling [51]. Similarly, numerous reports based on in vitro studies and glioblastoma tissues have documented the significant role of EGFR amplification and mutations in glioblastoma progression, although it is important to note that in 30–40% of glioblastomas, EGFR amplification does not occur [29,52], and EGFRvIII mutations are observed in only 20% of high-grade gliomas [52,53]. However, it has been indicated that poor prognosis low-grade gliomas (G2 grade) without IDH mutation are characterized by EGFR amplification [54]. To the best of our knowledge, there are no data indicating the direct involvement of kinins and their receptors in the activation of EGFR in glioma. Our genomic microarray data and validation results, showing higher *EGFR* mRNA as well as *MAP2K* mRNA levels in G2 glioma grade compared to G3 and G4 grade, provide new information regarding the genetic status of low-grade glioma in adults.

The current standard treatment has not been effective in preventing the progression of malignant glioma, as indicated by low survival rates. Standard therapy includes surgical resection followed by radiotherapy and chemotherapy [30,33]. However, chemotherapeutic agents have difficulty crossing the blood–brain barrier. In a malignant F98 glioma rat model, Cote et al. [55] demonstrated that B1R agonists induce selective blood-tumor barrier permeability through B1R activation and a COX-dependent pathway [55]. Later, these authors demonstrated (in the same model) that B1R and B2R agonists improved the delivery of the anticancer drug Carboplatin to the brain [56]. In fact, kinin analogs modulate blood–brain barrier permeability, making kinin receptor agonists a valuable tool for facilitating the transport of chemotherapeutic drugs [57].

The limitations of our study include a small sample size of patients with G3 stage tumors, which restricts the interpretation of study results. However, we were able to statistically compare the results between G2 and G3 as well as between G3 and G4 based on the evaluated number of tumor samples in each group. Another limitation of this study is the absence of well-known diagnostic and prognostic molecular biomarkers in glioma. Therefore, further prospective research on a more representative population is warranted to establish possible relationships between kinin receptor expression and other tumor molecular features, such as IDH mutation status, EGFR amplification, PTEN deletions/mutations, and MGMT methylation status. This comprehensive analysis could prove to be instrumental in both the diagnosis and the development of innovative treatment strategies for astrocytic brain tumors.

## 5. Conclusions

Our prospective study is one of the first to demonstrate the B1R and B2R patterns in patients with glioma. Nevertheless, further research is necessary to validate the potential role of kinin receptors in glioma treatment, particularly in the case of low-grade gliomas. This may have therapeutic significance, suggesting that kinin receptor agonists may enhance the transport of chemotherapeutic agents to brain glioma. Alternatively, blocking both kinin receptors simultaneously may decrease glioma growth and malignancy. In addition, to our knowledge, we have provided the first evaluation of kinin-related gene expression in tumor samples of patients with gliomas. The search for a biomarker that could select a molecular profile in patients with glioma is a challenging dilemma for researchers. Our results suggest that the assessment of mRNA *BDKRB1*, *PRKAR1A*, *MAP2K2*, and *EGFR* by RT-qPCR in samples of patients with gliomas may have diagnostic significance. Furthermore, the data we have presented could serve as a valuable reference for future investigations involving larger cohorts of patients. In these studies, it would be beneficial to explore the relationship between the expression of the mentioned genes and kinins receptors.

## Figures and Tables

**Figure 1 cancers-16-00241-f001:**
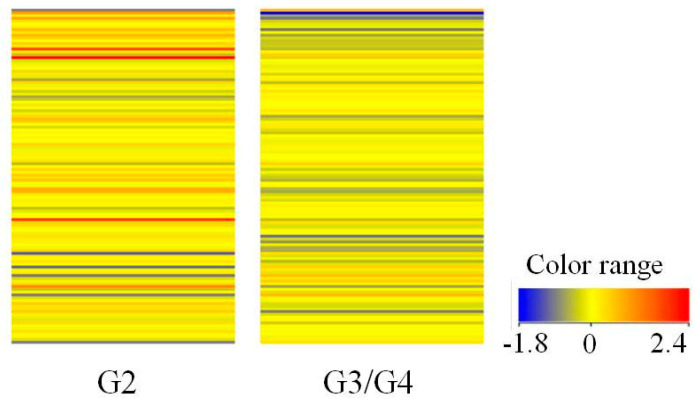
Graphical illustration of differences in expression profiles kinin-dependent genes based on tumor malignancy. The color variation between transcriptome groups indicates the presence of differences in gene expression profiles depending on tumor malignancy. Red—higher signal and high gene expression; blue—lower signal and low gene expression. G2 (*n* = 4) and G3/G4 (*n* = 6)—grade of tumor malignancy and number of samples (*n*).

**Figure 2 cancers-16-00241-f002:**
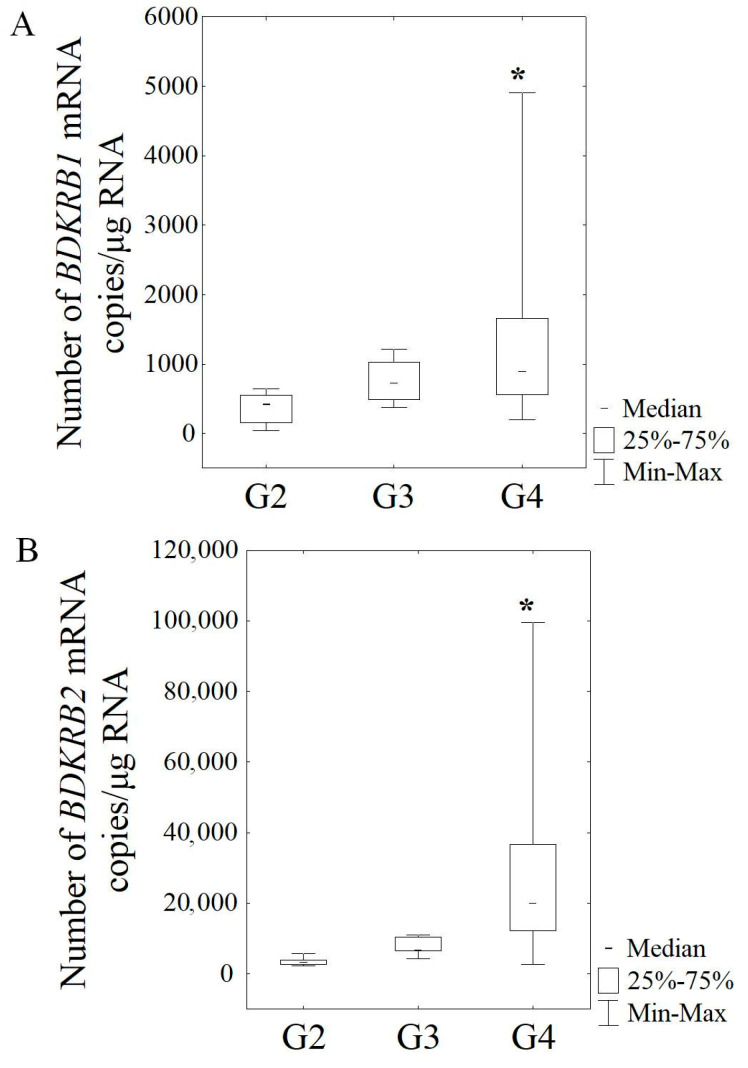
The number of *BDKRB1* (**A**) and *BDKRB2* (**B**) mRNA copies in patients with glioma. Results are presented as median with lower and upper quartiles and minimum and maximum values. G2 (*n* = 12), G3 (*n* = 5), and G4 (*n* = 26)—grade of tumor malignancy and number of samples (*n*); *—statistical significance (*p* < 0.05) vs. G2.

**Figure 3 cancers-16-00241-f003:**
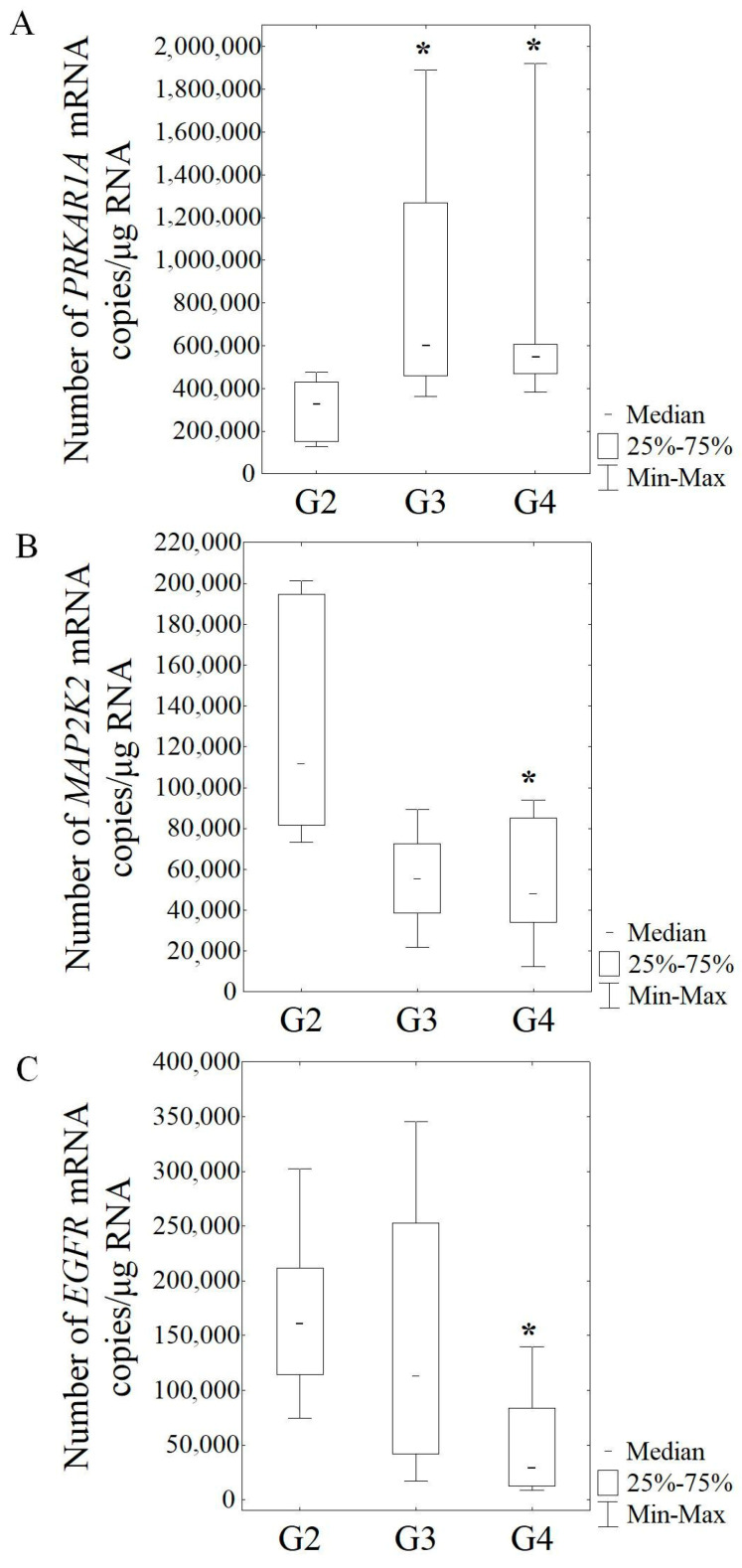
The number of *PRKAR1A* (**A**), *MAP2K2* (**B**), and *EGFR* (**C**) mRNA copies in patients with glioma. Results are presented as median with lower and upper quartiles and minimum and maximum values. G2 (*n* = 12), G3 (*n* = 5), and G4 (*n* = 26)—grade of tumor malignancy and number of samples (*n*); *—statistical significance (*p* < 0.05) vs. G2.

**Figure 4 cancers-16-00241-f004:**
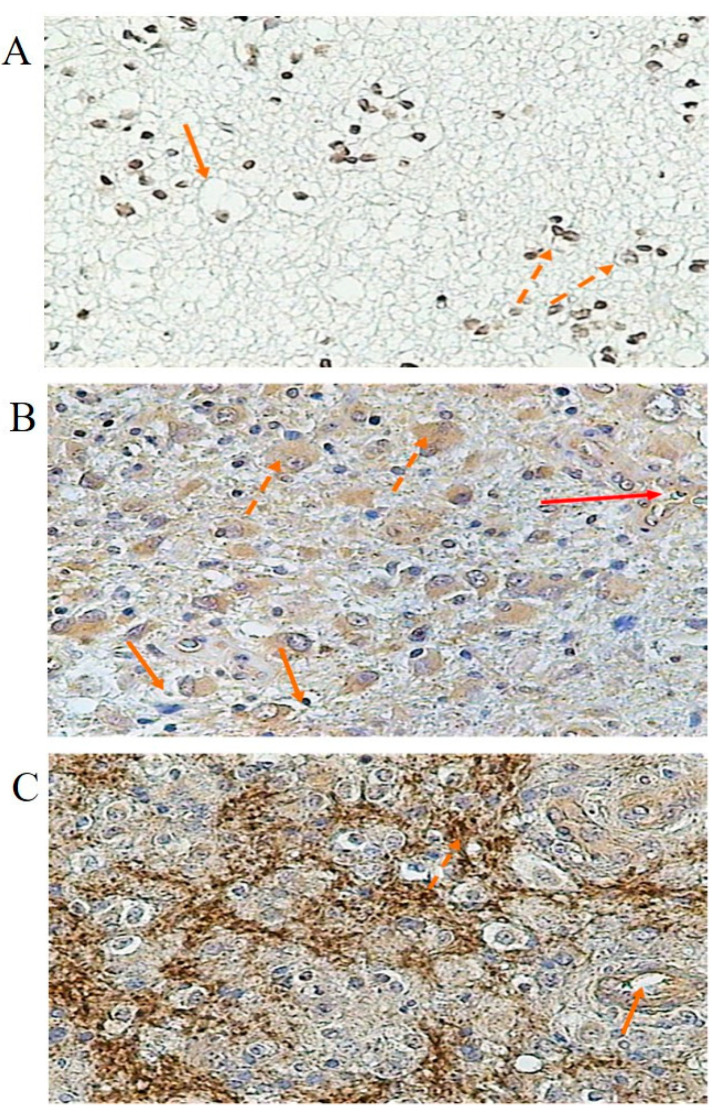
Specific staining reactions of BR1 protein in glioma of various stages. (**A**) G2—grade 2 of glioma; solid arrows: vascular endothelium; dashed arrows: glioma cells, slight reaction. (**B**) G3—grade 3 of glioma; solid arrows: endothelium; dashed arrows: glioma cells, intracellular reaction. (**C**) G4—grade 4 of glioma; solid arrows: cross-section of angiogenic vessel, dashed arrows: glioblastoma cells.

**Figure 5 cancers-16-00241-f005:**
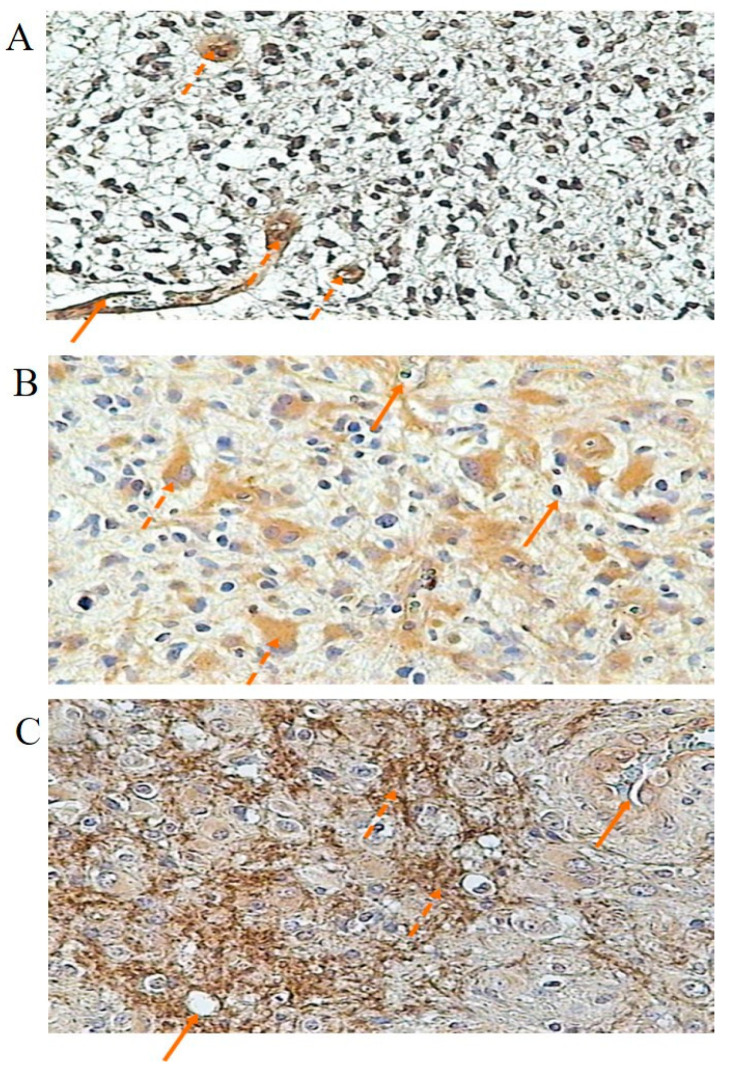
Specific staining reactions of BR2 protein in glioma of various stages. (**A**) G2—grade 2 of glioma; solid arrows: vascular endothelium; dashed arrows: staining in glioma cells. (**B**) G3—grade 3 of glioma; solid arrows: vessel and endothelial cell nucleus (at top of figure); dashed arrows: staining reaction in cytoplasm of glioma cell, and glioma cell with projections (in upper part of figure). (**C**) G4—grade 4 of glioma; solid arrows: cross-section through vessel and longitudinal section through the vessel (in upper part of figure); dashed arrows: glioblastoma cells; dashed arrows: capillary infiltrating cells (lower part).

**Table 1 cancers-16-00241-t001:** Primers used in the RT-qPCR reaction.

Gene	Oligonucleotide Sequence	Amplimer Length (bp)	Tm (°C)
*MAP2K2*	Forward: 5′ CTGGACTATATTGTGAACGAG 3′ Reverse:5′ CTTGATGAAGGTGTGGTTTG 3′	147	85.0
*SHC1*	Forward: 5′ GAGGAGAAAGCCCTGTAG 3′ Reverse: 5′ AGACGGTGAGAGTGATTG 3′	89	82.0
*EGFR*	Forward: 5′ GGAAAAGAAAGTTTGCCAAG 3′ Reverse: 5′ ATGAGGACATAACCAGCC 3′	195	81.4
*PRKACA*	Forward: 5′ CCAAAGTGTGTGGCAAAG 3′ Reverse: 5′ TCAGACTGGTCTATGTTAGC 3′	109	82.0
*PRKAR1A*	Forward: 5′ TTTAGAGTCTCTGGACAAGTG 3′ Reverse: 5′ TAATGAAGAACTCATCCCCTG 3′	119	82.2
*BDKRB1*	Forward: 5′ CCTTCATTTTCTGCCTGAG 3′ Reverse: 5′ GGAGAATCGTTTAAGCCTG 3′	105	84.2
*BDKRB2*	Forward: 5′ ACTTAGAAAAGCAAAGGGTG 3′ Reverse: 5′ CAATACTGATTCTGCTCCAC 3′	126	79.4
*ACTB*	Forward: 5′TCACCCACACTGTGCCCATCTACGA 3′ Reverse: 5′CAGCGGAACCGCTCATTGCCAATGG 3′	295	88.4

bp—base pairs. Tm—melting point. Forward—sense primer. Reverse—antisense primer.

**Table 2 cancers-16-00241-t002:** The number of probes differentiating the studied groups depending on statistical assumptions.

		Total *p*	*p* < 0.05	*p* < 0.02	*p* < 0.01	*p* < 0.005	*p* < 0.001
G3/4 vs. G2							
Number of probes		22,283	6378	3991	2760	1856	670
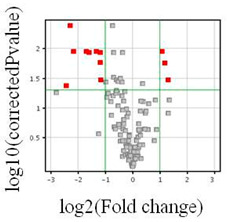	FC > 1.0	120	23	12	2	2	0
	FC > 1.1	93	23	12	2	2	0
	FC > 1.5	34	18	11	2	2	0
	FC > 2.0	16	12 *	9	1	1	0
	FC > 3.0	6	5	4	1	1	0

FC—fold change; *—statistical significance (*p* < 0.05), FC > 2.0. G2 and G3/G4—grade of tumor malignancy. The volcano plot—red squares are significant differentially expressed genes, grey squares are nonsignificant genes.

**Table 3 cancers-16-00241-t003:** Characterization of genes showing altered expression in gliomas of G3/G4 grade and G2 grade.

Probe Number	Gene Symbol	Name of Gene	FC	*p* *	Alteration of Expression G3/G4 vs. G2
200603_at	*PRKAR1A*	Protein Kinase CAMP-dependent Type I Regulatory Subunit Alpha	2.45	0.005	** ↑ **
213093_at	*PRKCA*	Protein Kinase C Alpha	2.25	0.002	** ↑ **
212688_at	*PIK3CB*	Phosphatidylinositol-4,5-Bisphosphate 3-Kinase Catalytic Subunit Beta	2.11	0.001	** ↑ **
211607_x_at	*EGFR*	Epidermal Growth Factor Receptor	5.46	0.008	** ↓ **
201469_s_at	*SHC1*	SHC Adaptor Protein 1	4.90	0.000	** ↓ **
200604_s_at	*PRKAR1A*	Protein Kinase CAMP-dependent Type I Regulatory Subunit Alpha	4.54	0.001	** ↓ **
213490_s_at	*MAP2K2*	Mitogen-activated Protein Kinase 2	3.25	<0.001	** ↓ **
200744_s_at	*GNB1*	G-protein Subunit Beta 1	3.05	0.001	** ↓ **
200852_x_at	*GNB2*	G-protein Subunit Beta 2	2.53	0.001	** ↓ **
208351_s_at	*MAPK1*	Mitogen-activated Protein Kinase 1	2.30	0.001	** ↓ **
201040_at	*GNAI2*	G-protein Subunit Alpha I2	2.27	0.002	** ↓ **
214853_s_at	*SHC1*	SHC Adaptor Protein 1	2.25	0.005	** ↓ **

G2 and G3/G4—grade of glioma malignancy; FC—fold change; *p* *—statistical significance; ↑—gene overexpression; ↓—gene down expression.

**Table 4 cancers-16-00241-t004:** Optical density of BR1 and BR2 proteins within astrocytic brain gliomas.

Grade of Malignancy	Number of Samples	BR1	BR2
G2	12	77.72 ± 7.36	70.05 ± 11.38
G3	5	105.10 ± 13.17 ^a^	116.79 ± 12.36 ^a^
G4	26	145.55 ± 15.13 ^b,c^	148.12 ± 19.12 ^b,c^

^a^—statistical significance, *p* < 0.05, G3 vs. G2. ^b^—statistical significance, *p* < 0.05, G4 vs. G2. ^c^—statistical significance, *p* < 0.05, G4 vs. G3.

## Data Availability

The data presented in this study are available in this article.

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
