# Peer review of "Kinin Receptors and Kinin-Related Gene Expression in Astrocytic Brain Tumors"

_cancers, 2024, doi:10.3390/cancers16020241_

Round 1

Reviewer 1 Report

Comments and Suggestions for Authors

The authors of this paper studied the biological role of the kinins and kinin receptors in the development of human astrocytic gliomas. They used samples from 43 patients with different tumor grades. The results obtained with the oligonucleotide microarray, RT-qPCR and immunohistochemical methods. The authors confirmed the involvement of kinins and their receptors in the pathogenesis of gliomas and identified kinin-dependent genes with altered expression patterns, which have potential as a diagnostic and therapeutic targets.

The paper is well written, the text clear and easy to read. The experimental design is appropriate to test the hypothesis. The statistical analysis are appropriate. The ethics statements and data availability statements are adequate. The statements and conclusions drawn coherent and supported by the citations. The references are appropriate and adequate. This is an interesting and important study that is relevant in this area. This article can be accepted for publication after minor revision.

1. On Figure 1, Table 2 and 3 data for G3 and G4 are combine. Were G3 and G4 data analyzed separately and no statistical differences were found between them? If this is true, it would be correct to add it to the text. If this is not true, this combination needs to be explained

2. On lines 449 – 466 the difference between G2 and G4 is discussed. What about G3?

3. The pictures contain text in a very small font that is difficult to read. I recommend doing them in a larger font.

4. In Figure 2 and 3, I recommend indicating which groups differ from each other

Author Response

Reviewer’s Comments to Author:

Review 1:

Comments:

The authors of this paper studied the biological role of the kinins and kinin receptors in the development of human astrocytic gliomas. They used samples from 43 patients with different tumor grades. The results obtained with the oligonucleotide microarray, RT-qPCR and immunohistochemical methods. The authors confirmed the involvement of kinins and their receptors in the pathogenesis of gliomas and identified kinin-dependent genes with altered expression patterns, which have potential as a diagnostic and therapeutic targets.

The paper is well written, the text clear and easy to read. The experimental design is appropriate to test the hypothesis. The statistical analysis are appropriate. The ethics statements and data availability statements are adequate. The statements and conclusions drawn coherent and supported by the citations. The references are appropriate and adequate. This is an interesting and important study that is relevant in this area. This article can be accepted for publication after minor revision.

Answer:

Dear Reviewer, we appreciate your highly positive feedback. We have made efforts to address all of the Reviewer's comments in order to enhance the quality of our work. We believe that the current version of the manuscript effectively describes the topic and has the potential for publication.

Comments:

1.On Figure 1, Table 2 and 3 data for G3 and G4 are combine. Were G3 and G4 data analyzed separately and no statistical differences were found between them? If this is true, it would be correct to add it to the text. If this is not true, this combination needs to be explained.

Answer:

The research utilized microarray analysis, a highly efficient but rather expensive method. Consequently, we regarded it as a screening method, and the analysis was conducted at the outset of the experiment, rather than on all tested samples. It's important to clarify that we combined the G3 and G4 groups in the analysis due to the limited number of G3 samples available, which was partly influenced by the frequency of occurrence of each type. Glioblastoma multiforme (G4 according to WHO) accounts for approximately half of all adult gliomas, while (G3 according to WHO) constitutes roughly 2.10% of all brain cancer cases. Additionally, we identified two distinct groups of transcriptomes through cluster analysis, revealing similar expression patterns: grade 2 (G2) gliomas and grade 3 and 4 gliomas (G3/G4). This categorization, with low-grade gliomas labeled as G2 and high-grade gliomas as G3/G4, served as the foundation for the subsequent microarray analysis, as explained in Section 3.1, “Gene expression profile of kinin-related genes based on oligonucleotide microarray sections.”

Comments:

  1. On lines 449 – 466 the difference between G2 and G4 is discussed. What about G3?

Answer:

We appreciate the reviewer's question. Indeed, we inadvertently omitted stage G3 from this section of the discussion. We have now made the necessary adjustments to rectify this omission.

We thank the reviewer for this question. Indeed, we have omitted stage G3 in this fragment of the discussion. We have made appropriate modifications to this indicated fragment.

Comments:

  1. The pictures contain text in a very small font that is difficult to read. I recommend doing them in a larger font.

Answer:

In accordance with Reviewer’s suggestion we modified the figures to make their more clear and visible.

Comments:

  1. In Figure 2 and 3, I recommend indicating which groups differ from each other.

Answer:

We would like to explain that in Figure 2 and 3 statistical significance * relate to the comparison vs. G2. In accordance with Reviewer’s suggestion we added the missing information in captions of Figure 2 and 3.

Reviewer 2 Report

Comments and Suggestions for Authors

Comments and Suggestions for the authors of the mabuscriot-cancers2763124-v1:

In this manuscript, the authors investigated the role of kinins and kinin receptors in astrocytic brain tumors, discussed the novelty of assessing BDKRB1 and BDKRB2 gene expression and B1R and B2R protein levels in human gliomas, and highlighted findings in the diagnosis of astrocytic brain tumors with potential implications for oncology and personalized treatment.

Suggestion:

·         In the Introduction section, the authors should provide a detailed discussion on the functional mechanisms underlying the involvement of kinins and their receptors in glioma progression, considering cell migration, invasion, angiogenesis, and pro-inflammatory mediator activation.

·         In the Result section, 1) the authors should provide the number of cases, especially in the figure legends, in which a p value exists, such as p<0.05, Gb3 vs Gb2, n=xx or Gb3(n=x1) vs Gb2(n=x2).

2) Since the authors have done the microarray study, it’s possible to provide and compare some import gene expressions with kinins and their receptors in this study.

·         In the Discussion section, the authors should consider expanding the scope of the study by exploring the correlation between kinin receptor expression and other tumor molecular features, such as IDH mutation status, EGFR amplification, PTEN deletions/mutations, and MGMT methylation status.

Author Response

Review 2:

Comments:

In this manuscript, the authors investigated the role of kinins and kinin receptors in astrocytic brain tumors, discussed the novelty of assessing BDKRB1 and BDKRB2 gene expression and B1R and B2R protein levels in human gliomas, and highlighted findings in the diagnosis of astrocytic brain tumors with potential implications for oncology and personalized treatment.

Answer:

Dear Reviewer, I would like to express my gratitude for your assessment of our work.

Comments:

In the Introduction section, the authors should provide a detailed discussion on the functional mechanisms underlying the involvement of kinins and their receptors in glioma progression, considering cell migration, invasion, angiogenesis, and pro-inflammatory mediator activation.

Answer:

We would like to express our appreciation for your suggestions. We have incorporated the suggested mechanisms and made modifications to the introduction section accordingly.

Comments:

In the Result section, 1) the authors should provide the number of cases, especially in the figure legends, in which a p value exists, such as p<0.05, Gb3 vs Gb2, n=xx or Gb3(n=x1) vs Gb2(n=x2).

Answer:

In accordance with Reviewer’s suggestion we added the missing information in figure legends.

Comments:

2) Since the authors have done the microarray study, it’s possible to provide and compare some import gene expressions with kinins and their receptors in this study.

Answer:

We would like to clarify that in our study, we assessed the expression profile of kinin-related genes, specifically 120 mRNA IDs. These genes were selected from a pool of 22,283 mRNA IDs based on data from the Affymetrix database and relevant literature. The assessment was conducted using oligonucleotide microarrays. Subsequently, when comparing the transcriptomes of glioma cells at various stages of advancement, we identified statistically significant alterations in the expression of 12 mRNA IDs. These differentially expressed genes are presented in Table 2.We greatly appreciate your observation, and we agree that comparing the expression of important genes related to kinins and their receptors may provide valuable insights for future studies. To address this, we have included this aspect in the limitations of our study and outlined it as a potential focus for future research.Thank you for bringing this to our attention.

Comments:

In the Discussion section, the authors should consider expanding the scope of the study by exploring the correlation between kinin receptor expression and other tumor molecular features, such as IDH mutation status, EGFR amplification, PTEN deletions/mutations, and MGMT methylation status.

Answer:

We thank the Reviewer for this valuable suggestion. We have incorporated a dedicated section in the discussion of our research, highlighting the potential for future exploration of the correlation between kinin receptor expression and other molecular features of cancer.Thank you for your input.

Reviewer 3 Report

Comments and Suggestions for Authors

In this study the authors investigated 43 patients with gliomas (twelve grade G2, five with G3 and 26 with G4) for the expression of kinin receptors and kinin-dependent genes. They found that kinins and their receptors are involved in the development of glioma.

The authors address a very important issue in the research of astrocytic brain tumors. The applied methods are well described and results are well written and good illustrated. This makes the paper easy to follow.

Although I´m not convinced that the B1R and B2R patterns may have therapeutic impact it is the first time, at least for my knowledge, that kinik-related gene expression in in gliomas were evaluated.

Especially the discussion part is very detailed and the most important literature is cited. This makes the paper very attractive for the readers of Cancer.

Author Response

Review 3:

Comments:

In this study the authors investigated 43 patients with gliomas (twelve grade G2, five with G3 and 26 with G4) for the expression of kinin receptors and kinin-dependent genes. They found that kinins and their receptors are involved in the development of glioma.

The authors address a very important issue in the research of astrocytic brain tumors. The applied methods are well described and results are well written and good illustrated. This makes the paper easy to follow.

Although I´m not convinced that the B1R and B2R patterns may have therapeutic impact it is the first time, at least for my knowledge, that kinin-related gene expression in in gliomas were evaluated.

Especially the discussion part is very detailed and the most important literature is cited. This makes the paper very attractive for the readers of Cancer.

Answer:

We would like to extend our heartfelt appreciation to the Reviewer for their high assessment of our manuscript. We are delighted that our discussion has been well received and that the significance of the issue we have addressed, which is relatively novel, has been recognized.We have made several additional minor corrections to the article to further enhance its overall quality.Regarding the patterns of B1R and B2R and their potential therapeutic implications, we believe that this avenue of research holds great promise for the future. Kinins, by stimulating B1R and B2R, as well as nitric oxide and prostaglandins released from vascular endothelium, can enhance blood vessel permeability, thereby facilitating metastasis, promoting angiogenesis, and accelerating nutrient transport to tumor cells. Consequently, the use of selective kinin receptor antagonists may indeed have a beneficial therapeutic impact.Once again, we would like to express our gratitude for your meticulous review of our manuscript and your invaluable assessment.

Reviewer 4 Report

Comments and Suggestions for Authors

This study demonstrates the B1R and B2R patterns in patients with glioma, and it holds significant interest and potential for impact. Additionally, it is a prospective study. However, I believe it is unnecessary to include a simple summary and that the abstract does not present a clear exposition of the results obtained in the study. In the introduction, only a small portion of the objective is explained. Furthermore, the discussion is very long and the novelties introduced by this article are not highlighted compared to previous studies. The significance of the isocitrate dehydrogenase (IDH) mutation status for Low Grade Gliomas is ambiguous in the discussion section, and it is not explicitly stated that it is an independent prognostic factor for survival (doi:10.21873/anticanres.16432). I also believe it would be more advantageous to incorporate a section on limitations into the study. I would recommend that the authors edit lines 479-481 to indicate that further studies are needed to evaluate the possible role of the kinin receptor in the treatment of low-grade gliomas.

Author Response

Review 4:

Comments:

Does the introduction provide sufficient background and include all relevant references – Can be improved

Are all the cited references relevant to the study – Can be improved

Is the research design appropriate – Can be improved

Are the methods adequately described – Can be improved

Are the results clearly presented – Can be improved

Are the conclusions supported by the results – Can be improved

Answer:

We are thankful for the Reviewer's valuable suggestions, which led to the enhancement and improvement of the indicated sections in our work. We have made diligent efforts to address all of the Reviewer's comments in order to enhance the overall quality of our research.

Comments:

This study demonstrates the B1R and B2R patterns in patients with glioma, and it holds significant interest and potential for impact. Additionally, it is a prospective study. However, I believe it is unnecessary to include a simple summary and that the abstract does not present a clear exposition of the results obtained in the study. In the introduction, only a small portion of the objective is explained. Furthermore, the discussion is very long and the novelties introduced by this article are not highlighted compared to previous studies. The significance of the isocitrate dehydrogenase (IDH) mutation status for Low Grade Gliomas is ambiguous in the discussion section, and it is not explicitly stated that it is an independent prognostic factor for survival (doi:10.21873/anticanres.16432). I also believe it would be more advantageous to incorporate a section on limitations into the study. I would recommend that the authors edit lines 479-481 to indicate that further studies are needed to evaluate the possible role of the kinin receptor in the treatment of low-grade gliomas.

Answer:

We sincerely appreciate your feedback and your positive comments regarding our research. It is heartening to know that you recognize the significance of our work.

Regarding the modifications, we have indeed made changes to the abstract and introduction sections, as you suggested. We have also noted your observation about the discussion section's length and would like to clarify that we didn't shorten it because other Reviewers had no objections to this section. Also, the “Simple Summary" section hasn't been removed due to it being a requirement in the Journal of Cancers.

We fully agree with your point that further studies are necessary to assess the potential role of kinin receptors in the treatment of low-grade gliomas. Consequently, we have incorporated the following sentence into our manuscript, in accordance with your suggestion: "Nevertheless, further research is necessary to validate the potential role of kinin receptors in glioma treatment, particularly in the case of low-grade gliomas." Additionally, we have highlighted some of the limitations of our research.

We hope that our article, in its current form, will meet the acceptance criteria and be deemed suitable for publication.

Round 2

Reviewer 4 Report

Comments and Suggestions for Authors

I appreciate the changes you have made to your article and appreciate you taking my suggestions into consideration.